# Carbon Aerogels as Electrocatalysts for Sustainable Energy Applications: Recent Developments and Prospects

**DOI:** 10.3390/nano12152721

**Published:** 2022-08-08

**Authors:** Minna Zhang, Xiaoxu Xuan, Xibin Yi, Jinqiang Sun, Mengjie Wang, Yihao Nie, Jing Zhang, Xun Sun

**Affiliations:** 1Shandong Key Laboratory for Special Silicon-Containing Material, Advanced Materials Institute, Qilu University of Technology (Shandong Academy of Sciences), Jinan 250014, China; 2Key Laboratory of High Efficiency and Clean Mechanical Manufacture, Ministry of Education, School of Mechanical Engineering, Shandong University, Jinan 250061, China

**Keywords:** carbon aerogels, electrocatalysis, sustainable energy applications

## Abstract

Carbon aerogel (CA) based materials have multiple advantages, including high porosity, tunable molecular structures, and environmental compatibility. Increasing interest, which has focused on CAs as electrocatalysts for sustainable applications including oxygen reduction reaction (ORR), oxygen evolution reaction (OER), hydrogen evolution reaction (HER), and CO_2_ reduction reaction (CO_2_RR) has recently been raised. However, a systematic review covering the most recent progress to boost CA-based electrocatalysts for ORR/OER/HER/CO_2_RR is now absent. To eliminate the gap, this critical review provides a timely and comprehensive summarization of the applications, synthesis methods, and principles. Furthermore, prospects for emerging synthesis, screening, and construction methods are outlined.

## 1. Introduction

The energy released by burning fossil fuels is still the primary energy needed today. While the burning of fossil fuels causes environmental pollution, people will inevitably face the problem of the depletion of fossil fuels [1,2]. The mutual conversion of electrical energy and chemical energy by utilization of renewable energy is considered a feasible method to alleviate the crisis of fossil fuels [3]. The conversion between electrical and chemical energy requires a large amount of energy because it generally involves breaking or forming chemical bonds between molecules such as water (H_2_O), hydrogen (H_2_), oxygen (O_2_), and carbon dioxide (CO_2_) [4,5,6,7,8,9]. Therefore, it is essential to design catalysts with high activity, selectivity, and stability. 

In recent decades, nanostructured electrocatalysts with large pores, diverse compositions, and controlled structures have been popularly researched. Among these, carbon materials are widely used in electrocatalysis due to their extraordinary electrical conductivity and excellent mechanical and chemical properties [10,11,12]. However, impurities and difficult-to-control structures in natural carbon are detrimental to the application of high-quality materials. Since Pekala first obtained a layered three-dimensional (3D) porous network structure of resorcinol formaldehyde (RF) carbon aerogel (CA) in 1989 [13], the numerous research on CAs with customizable physical structures and chemical properties provides more possibilities for the application of carbon materials [14,15]. CAs are conventionally formed by the carbonization of aerogels in which gas accounts for more than 90% of the volume of the entire structure. This absorbent gel maintains a highly interconnected 3D porous network with solid structural stability [16]. The micropore and mesopore formations of CAs are independently controlled, which has obvious advantages for preparing porous materials [17]. CAs are endowed with various unique properties by their extremely high porosity, such as extremely high porosity (up to 99.8%), extremely high specific surface area (up to 2000 m^2^ g^−1^), rich pore structure (pore size distribution in 1–100 nm), and very low density (as low as 3 mg cm^−3^). These unique properties have led to the development of CAs in energy storage, adsorption, sensing, and thermal insulation in addition to their applications in electrocatalysis [18,19,20].

The preparation of CAs generally involves three steps: gelation, drying, and carbonization (Figure 1a). Firstly, gels composed of liquid phases in a 3D solid network are synthesized using various gel techniques, such as the sol-gel methods, freeze–thaw-induced gels, or polymer cross-linking. Gels are generally divided into two categories: hydrogels (the liquid phase is a gel of water) and organogel (the liquid phase is a gel of organic solvents). Synthesis of metal-supported CA catalysts often introduces metal ions and heteroatom into the aerogel system during this step [21]. Secondly, the prepared gel is dried to remove the liquid within the gel network. Drying methods are generally divided into three types: supercritical drying, freeze-drying, and atmospheric drying [22,23,24]. The liquid inside the gel is replaced by air through special drying techniques, such as atmospheric drying, freeze-drying, and supercritical drying, to obtain aerogels that can well maintain the original gel structure. Finally, CA is obtained by carbonizing the aerogel obtained after drying. Traditional CAs are obtained by carbonization of organic aerogels such as RF aerogels. Most of these CAs are inflexible and brittle solids, and the microstructure is challenging to control and the surface is not easy to modify [25]. With the advancement of research, nano-aerogel materials that are simpler to synthesize, greener, and more easily adjustable, such as biomass, graphene, and carbon nanotubes, are being developed in large numbers [11,26,27,28,29,30,31,32]. New properties frequently emerge during the gelation or assembly of these nanostructures. Table 1 summarizes the classification and characteristics according to the precursor types of CAs. The resulting CA catalysts offer ideal combinations of highly exposed reaction sites, self-supporting frameworks, conductivities, and direct mass transfer pathways.

Given the rapid prosperity of the CA electrocatalysts, we believe a critical review of their applications, synthesis, and prospects is warranted. In this review, the latest applications of CAs in (1) oxygen reduction reaction (ORR), (2) water splitting evolved hydrogen evolution reaction (HER) and oxygen evolution reaction (OER), and (3) electrocatalytic CO_2_ reduction reaction (CO_2_RR) are first reviewed and compared. Then, the factors influencing the performance of heteroatom doping and metal-anchored CAs as electrocatalysts are analyzed. Finally, prospects for the existing challenges and future developments are discussed.

## 2. Applications of CAs in Electrocatalysis

Three-dimensional interconnected porous nanostructured CAs have been widely used in electrocatalysis due to their excellent electrical conductivities, large specific surface areas, and designable structures [34]. The large specific surface area of the 3D structure exposes the possible active sites, which enhances the contact between guest molecules (CO_2_, O_2_, H_2_O) and the active sites, thus improving the catalysis rate of the reaction. The hierarchically porous structure ensures ultrafast mass transfer between the catalyst and guest molecules [35]. The macroporous structures are conducive to the diffusion of gas and electrolytes on a micro-scale; meanwhile, the micropores help improve the transportation rate of the gas molecules to the active sites on a nano-scale. Furthermore, the tightly connected 3D superstructures between the frameworks enable electrons to transfer as fast as a car on the highway to obtain good electrical conductivity [36,37].

Due to these properties possessed by CAs, they are designed as catalyst supports and assembled to obtain highly active and stable electrocatalysts that combine the properties of aerogels with those of nano components (such as catalytic activity and electrical conductivity). In constructing CAs, heteroatom doping (such as B, S, N, and F) can significantly improve the electrochemical activity (Figure 2a) [38,39]. Doping heteroatoms with different electronegativity into the carbon network introduces asymmetric charges, redistributes the spin density, and destroys the electroneutrality of the carbon matrix to optimize the carrier concentration and obtain more catalytically active sites. Furthermore, additional catalytically active species (such as metal NPs or metal oxide, etc.) can be loaded on the carbon aerogel surface to tune the surface chemistry (Figure 2a) [40]. Metal nanoparticles have been widely studied in various morphologies, compositions, and structures as efficient electrocatalysts due to their high activity [41,42]. In most cases, metal nanoparticles are loaded on carbon materials, which rely on the large specific surface area of carbon materials to fully expose the active center sites (Figure 2b) (CAs generally have higher specific surface areas than other carbon materials). The stability of CAs at high voltage is also one of the critical factors in ensuring catalyst activity.

In addition, emerging CAs catalysts formed from carbon materials with good electrical conductivity, such as carbon nanotubes, graphene, and carbon nanofibers, have also attracted considerable attention in electrocatalysis [43]. For reactions involving gases, practical applications ultimately require electrocatalysts as gas diffusion electrodes. The self-supporting nature of the carbon aerogel results in direct contact of the electroactive species with the conductive substrate, thus ensuring the excellent integrity of the catalyst layer.

In most electrocatalytic reactions, the CAs can be used both as catalysts and catalyst supports. In this section, three types of typical electrocatalytic reactions (Figure 1b) with CAs employed are compared in detail, CAs synthesis methods are discussed, and the design principles of CAs-based electrocatalysts are presented.

### 2.1. ORR

The slow kinetics and high overpotential of the ORR occurring at the cathode in metal–air battery and fuel cell technologies have been important factors limiting its development. According to previous studies, the ORR can be divided into two types: reactions with four-electron transfer or two-electron transfer [44,45]. The four-electron transfer reactions are: O_2_ + 2H_2_O + 4e^−^ → 4OH^−^ (alkaline medium), O_2_ + 4H^+^ + 4e^−^ → 2H_2_O (acidic medium). The two-electron transfer reactions are: O_2_ + H_2_O + 2e^−^ → HO_2_^−^ + HO^−^ (alkaline medium), O_2_ + 2H^+^ + 2e^−^ → H_2_O_2_ (acidic medium). Obviously, a catalyst that promotes the four-electron transfer reaction is what we need. Currently, platinum (Pt)-based catalysts are still the best catalysts for ORR, usually synthesized by dispersing Pt nanoparticles onto carbon supports to solve the problem of the reduced exposed active surface due to aggregation and non-uniform dispersion of platinum nanoparticles [46,47,48,49]. Highly active Pt/C catalysts have been commercialized. However, the scarce Pt resources, high cost, and poor durability are the keys to limiting their wide application. The search for efficient alternative catalysts is inevitable. We pursue catalysts with an excellent four-electron process, low H_2_O_2_ yield, satisfactory half-wave potential, strong toxicity tolerance, and better stability than commercial Pt/C. Figure 3 and Table 2 summarizes the ORR performance in recent years of CA material catalysts with a four-electron pathway. Low-cost and high-efficiency heteroatom-doped metal-free carbon-based catalysts and carbon-supported non-noble metal-based catalysts are expected to be ideal substitutes for Pt-based catalysts.

#### 2.1.1. Heteroatom-Doped Metal-Free CAs

Catalytically active carbon materials are promising metal-free catalysts to replace noble metal catalysts. They are more stable and resistant to toxicity than metal-based materials. Although carbon materials have minimal catalytic properties, carbon materials modified by doping with nonmetallic heteroatoms (B, N, S, P, and I, etc.) are a promising class of electrocatalysts for ORR [54]. It can even match the ORR catalytic performance of commercial Pt/C in an alkaline environment. The incorporation of heteroatoms affects the changes in the spin density and charge of nearby C atoms, endows doped carbon with metallic properties, and enhances the catalytic performance of carbon materials. Since the discovery of N-doped carbon nanotubes as an efficient carbon-based metal-free electrocatalyst for ORR in 2009, carbon-based metal-free catalysts have been extensively studied and used in electrocatalysis [55]. As one of the most common heteroatoms, N is often used to enhance the electrochemical activity of ORR because its atomic size is similar to that of carbon, forming solid covalent bonds between N and surrounding carbon atoms. Krisztina László et al. [56]. introduced melamine (M) as an N source, with resorcinol (R) and formaldehyde (F) used as raw materials, and prepared 3D mesoporous CAs by adjusting the M/R ratio during the polymerization process. N mainly exists on the surface in the structure of pyridine-N and in the form of condensed polycyclic aromatic hydrocarbon systems, in which N replaces C and O exists in the form of single and double bonds with carbon atoms. At an N content of 4%, the onset potential of the monotonic growth of ORR is close to that of polyacrylonitrile-derived carbon. The strong correlation between the BET surface area and the cathode current density reveals the role of the accessibility of the N species in the electrocatalytic process.

In addition to the electrochemical activity, the large specific surface area (i.e., more active sites) of heteroatom-doped carbon catalysts is also a critical parameter affecting ORR’s high performance. The self-supporting structure of 3D aerogels can effectively prevent the reduction of the effective surface area of carbon materials caused by π–π repacking in heteroatom doping. The unique aerogel structure with a layered structure, large porosity, extended surface area, and highly exposed active sites is beneficial for enhancing ORR performance. For example, surface-modified CAs and graphite oxide (GO) composites with S atoms exhibited good ORR catalytic activity in alkaline media and excellent tolerance to methanol crossover. Aerogel-like N-doped carbon (NDC−MS) with an extensive surface area (1548.6 m^2^ g^−1^) and relatively high pore volume were synthesized using the molten salt method (MS) [57]. The synthesized NDC−MS showed excellent ORR catalytic activity, with a half-wave potential as high as 0.88 V. Lai et al. [58] reported a biomass-derived method to obtain P-doped carbon dots/graphene (P−CD/G) nanocomposites. The P−CD/G nanocomposite exhibits excellent ORR activity, comparable to commercial Pt/C catalysts. When used as a cathode material for primary liquid aluminum–air batteries, the device with P−CD/G nanocomposite showed an impressive power density of 157.3 mW cm^−2^ (compared to a similar Pt/C cell of 151.5 mW cm^−2^) and a stable discharge voltage of ≈1.2 V under different bending states. In addition, the activation of CAs by KOH can increase the BET surface area and hierarchical porous structure (especially micropores). This process can significantly affect the content of reactive N species and oxygen functional groups (C–O–C and –COOH). Zhou et al. [50] found that the pyridine-N content increased with the increase in KOH, and the ORR activity was linearly positively correlated with the pyridine-N content (Figure 3a). The reduced C–O–C and –COOH contents in the CA activated by KOH ensured a lower H_2_O_2_ yield, thereby enhancing the ORR efficiency. When the mass ratio of KOH to CA was 10, the highest power density was 967 ± 34 mW m^−2^, which was 37.4%, 55.2%, and 283.7% higher than Pt/C, activated carbon, and CA, respectively. Its excellent performance is attributed to the superior BET surface area (1827 m^2^ g^−1^), numerous micropores, the highest pyridine-N content (0.26 at. %), the lowest pyrrolic-N, and reduced C–O–C and COOH content.

Furthermore, multiple heteroatom doping will produce synergistic effects, and such co-doped/multi-doped carbonaceous materials can possess more active sites than most single-doped carbonaceous materials. There are differences in electronegativity between heteroatoms (B (2.04), N (3.04), O (3.44), F (4.0), P (2.19), S (2.58), and C (2.55), and the synergistic interactions between heteroatoms promote ORR during co-doping or polyatomic doping. Lin et al. [59] co-doped N and F into carbon nanowire aerogels. The obtained N−F co-doped carbon nanowire aerogels (NFCNAs) exhibited an onset potential and half-wave potential of 0.912 and 0.825 V, respectively, with comparable ORR catalytic performance to Pt/C. Wang et al. [60] prepared a series of 3D porous B−N co-doped graphene aerogels (BN−GAs) using NH_4_B_5_O_8_ as the precursor of the B-N source. The BN-GAs show similar ORR activity but superior stability and superior methanol tolerance to Pt/C. Han et al. [61] obtained N, P double-doped porous carbon (NPPC) with a large specific surface area by CO_2_ activation of a novel N, and P-doped aerogel (NPA). CO_2_ activation, as a corrosion-free and environmentally friendly method, can not only generate abundant mesopores and micropores in NPPC but also positively influence the conversion of N species from pyrrolic-N to pyrrolic-N and graphitic-N, thereby promoting ORR activity. Hence, compared with carbonized NPA, NPPC shows higher ORR activity, with a more significant number of transferred electrons (3.95 at 0.2 V vs. HER). In addition, NPPC exhibits better long-term stability than 20 wt % Pt/C. Some biomass materials are rich in heteroatoms, which can be used as pyrolysis precursors for self-doped heteroatom carbon materials. Dong et al. [62] prepared N-S self-doping nanoporous carbon spheres (NSCs) derived from onion as efficient metal-free electrocatalysts (Figure 3b–e). It exhibits excellent catalytic activity towards ORR via the 4e^−^ mechanism, with an onset potential of 0.88 V (vs. RHE) and excellent stability compared to Pt/C catalysts. This high activity is not only attributed to the synergistic effect of S and N doping in carbon and sufficient active centers but also the catalyst’s suitable microporous structure and large specific surface area.

Recent studies have demonstrated that using heteroatom-incorporated CAs as ORR catalysts instead of noble metal catalysts is a low-cost and effective strategy. Notably, heteroatom-doped CAs typically catalyze ORR in alkaline media, limiting their application in fuel cells due to the potential for CO poisoning of the electrolyte. Researchers are now turning their eyes to the catalysts that stabilize reactions in acidic electrolytes. Novel metal-free materials with N–S–C coordination structure active sites constructed from N-modified S defects in CAs with 3D hierarchical macro-meso-microporous structures exhibiting excellent ORR activity, good stability, and high current density in alkaline and acidic electrolytes [51]. The half-wave potential was 0.76 V in 0.5 M H_2_SO_4_ and 0.1 M HClO_4_, and 0.85 V in 0.1 M KOH. The experimental and computational results show that the pentagonal S defect is the key to ORR in acidic electrolytes because it can significantly enhance reactivity. A graphitic-N at the site of the pentagonal defect element further significantly enhances the reactivity due to the precise local control of the electronic structure, thereby forming a highly active area for ORR in acid. Nevertheless, ORR in acid fuel cells still has a long way to go. Despite the more active areas of these CAs catalysts, it seems that highly loaded electrode preparation is required for good ORR performance.

**Table 2 nanomaterials-12-02721-t002:** Overview of the ORR performance of the obtained four-electron pathway for different CA electrocatalysts.

CA Materials	Dominant Active Species	S_BET_ [m^2^ g^−1^]	Electrolyte	Initial Potential [V_RHE_]	Half-Wave Potential [V_RHE_]	Power Densities [mW cm^−2^]	Ref.
Fe_3_C/NCA	Fe_3_C	274	0.1 M KOH	0.93	0.83	253	[63]
Bi–CoP/NP-DG	Bi–CoP	143	0.1 M KOH	0.93	0.81	122	[64]
Co/NCA	Co–Nx	422	0.1 M KOH	0.95	0.83	~244	[65]
FeNx-CN/g-GEL	FeNx	1040	0.1 M KOH	1.00	0.90	173	[66]
Fe-UP/CA	Fe-UPs	957	0.1 M KOH	1.08	0.93	140	[67]
NSC-A2	N–S	1558	0.1 M KOH	0.88	0.76	~320	[62]
BN-GAs	B–N	227	0.1 M KOH	−0.05	−0.20	~70	[60]
MCG-2	Mn–N	754	0.1 M KOH	0.988	0.859	112	[68]
Ce/Fe/NCG-x	Ce/Fe	699	0.1 M KOH	1.45	0.97	101	[52]
PTEBbpyCu-HT	Cu-byp	625	0.1 M KOH	0.86	0.68	~100	[69]
NSCA-700-1000	N–S–C	1307	0.1 M KOH	0.65	0.85	~18	[51]
			0.5 M H_2_SO_4_		0.76		
			0.1 M HClO_4_		0.76		
Co–N–GA	Co–N	485	0.5 M H_2_SO_4_	0.88	0.73	~100	[70]
			0.1 M KOH	0.99	0.85		
NFCNAs-18-1000	N, F-co doped	768	0.1 M KOH	0.91	0.83	/	[59]
NCAs-800	N–C	/	0.1 M KOH	0.85	0.80	1048	[44]
GA	porous effect	63	0.1 M KOH	−0.12	~−0.32	~148	[71]

#### 2.1.2. CAs as Non-Noble Metal-Based Support

The key to the breakthrough of non-noble metal electrocatalysts in ORR is to obtain performance comparable to that of noble metal catalysts [68,72]. Metal particles are easily sintered and aggregated at high temperatures, which is not conducive to the progress of catalytic reactions. To limit the behavior of metal particles, they are often fixed on carbon substrates. Unlike heteroatom-doped CAs, non-noble metal-modified carbon aerogel catalysts use carbon aerogel as a support, with non-noble metals anchored. Enhanced dispersion and stabilization of vibrant metal centers are the essences of non-noble metal electrocatalysts. Many studies have confirmed that doping non-noble metal particles into carbon aerogel matrices can expose as many active sites as possible while providing a stable environment. What is more, combining with the heteroatoms doped in carbon improves the stability and activity of the metal center and obtains ORR performance comparable to that of noble metal-based Pt/C.

Based on the previous study, Wang et al. [52] strategically utilized Ce/Fe elements to functionalize N-doped CAs. In this work, the N content of the pyridine type was increased from 16.71% to 20.28%, which, combined with the macroporous structure of CAs, effectively improved the ORR catalytic performance. The catalyst was tested for ORR, and the on-cell voltage reached 1.450 V, comparable to commercial Pt/C catalysts (Figure 3f–k). It is confirmed that adding heteroatoms to the carbon matrix can increase the interfacial charge transfer rate between the metal and the heteroatom-doped carbon and improve the ORR performance of the catalyst. Chen et al. [66] prepared a FeNx atomic cluster catalyst (FeNx–CN/g–GEL) supported by 3D porous N-rich graphene CAs. It exhibits excellent catalytic performance for ORR, with a half-wave potential of 0.90 V superior to commercial Pt/C (0.835 V) and a mass-specific activity of 840 mA mg_Fe_^−1^ at 0.80 V. It also has good long-term stability, with only a slight decrease after 28,000 s test, and, more importantly, significant tolerance to methanol and SCN^−1^.

Rechargeable metal–air batteries experience ORR and OER during discharge and charge, respectively, and it is inevitable to develop bifunctional catalysts for this purpose. Tang et al. [53] set a new hybrid hydrogel to prepare Co_9_S_8_-doped honeycomb CAs, as shown in Figure 3l. The excellent bifunctional activity and robust stability of the newly developed catalysts in ORR and OER are due to the remarkable synergy between 3D porous N, P-co-doped CAs, and Co_9_S_8_. Bimetallic catalysts have enhanced activity and stability compared to their monometallic counterparts. Chen et al. [73] used bimetallic hydrogel templates to prepare CAs containing Fe–Co bimetallic sites (NCAG/Fe–Co) as bifunctional electrocatalysts for ORR and OER. The ORR/OER potential difference of NCAG/Fe–Co was only 0.64 V, which was 60 mV lower than that of commercial Pt/C and RuO_2_ catalysts (0.70 V) at a current density of 10 mA cm^−2^. The bimetallic NCAG/Fe–Co catalyst was applied to flexible metal–air batteries. Moreover, it exhibits good rechargeability, flexibility, and excellent performance, with an open-circuit voltage of 1.47 V and a maximum power density of 117 mW cm^−2^. Xia et al. [74] prepared Fe–Ni alloy (Fe–Ni) nanoclusters (Fe–Ni ANCs@NSCA catalyst) anchored on N, S co-doped CAs by freeze-drying. The Fe–Ni ANC@NSCA catalyst is a bifunctional catalyst in an alkaline medium with a half-wave potential of 0.891 V in the ORR and an overpotential at 260 mV with a current density of 10 mA cm^−2^ in the OER, which outperforms existing Pt/C catalysts and RuO_2_ catalysts.

### 2.2. HER and OER

Hydrogen energy is a green and clean renewable energy with high combustion calorific value. The electrocatalytic water splitting reaction, driven by electricity generated from renewable energy sources such as photovoltaics, hydropower, and wind power, splits water to produce H_2_ and O_2_, which is one of the ways to use and convert renewable energy. Water electrolysis produces H_2_ and O_2_ in an environmentally friendly manner through two half-cell reactions [75], i.e., hydrogen evolution reaction (HER): 2H^+^_(aq)_ + 2e^−^
→ H_2(g)_; oxygen evolution reaction (OER): 2H_2_O_(l)_ → O_2(g)_ + 4H^+^_(aq)_ + 4e^−^. However, the efficiency of water electrolysis and its application to industrial production is greatly limited by the slow kinetics of the two half-reactions, HER and OER. Therefore, it is imperative to develop low-cost, high-activity catalysts (As shown in Figure 4).

In recent years, CA-based materials with large specific surface area and self-supporting structures have received special attention due to their excellent catalytic activity and durability in catalysis. As a carrier, CAs can effectively solve the problem of the reduction in or inactivation of the catalytic effect caused by the swelling and aggregation of nanoparticles during the reaction process. Robust 3D CAs with large specific surface areas and abundant open pores provide navigable channels for electrolyte contact and electron transfer and enable a very intimate connection between electrolyte and catalysts. Transition metal-based materials (such as transition metal carbides, transition metal sulfides, transition metal phosphide, etc.) have attracted much attention due to their chemical and physical properties, such as Pt-like electronic structures, high electrical conductivity, and broad pH stability. Loading them onto CAs is a feasible method to enhance the catalyst activity. Xie et al. [76] synthesized carbon core-shell materials (CoP@C−NPs/GA−x (x = 5, 10, 20) by embedding CoP nanoparticles into different graphene aerogels (GAs) to improve the poor electrochemical reaction kinetics of HER (Figure 4a–c). The synthesized CoP@C−NPs/GA-5 exhibited potent catalytic activity in acidic and alkaline electrolytes with small overpotentials of 120 and 225 mV and low Tafel slopes of 57 and 66 mV dec^−1^ at a current density of 10 mA cm^−2^, respectively. It is similar to the encapsulation of empty (Co, Fe) P nano-frames in N, P-co doped GAs (CFP NFs@NPGA) (Figure 4d–f) [77]. Elbaz et al. [81] prepared porous, high-surface-area molybdenum oxide aerogels by the sol-gel method and carburized with methane to develop a durable Pt group metal-free catalyst for hydrogen production. Compared with the Pt/C catalyst, the MoC aerogel exhibits remarkable stability, with an overpotential drift of only 10 mV after stability testing, compared with 100 mV for Pt/C. Transition metal phosphates with hollow structures are loaded onto carbon aerogel substrates to achieve efficient charge transfer and enhance OER activity. Guo et al. [4] prepared highly efficient and stable electrocatalysts and ZIF-67-derived hollow Ni–Co phosphate nanocages supported by waste paper-based CAs (NCP@WPCA). Benefiting from the enhanced electrochemically active surface area and fast gas emission, the catalyst exhibits excellent OER activity. NCP@WPCA exhibits good stability in alkaline solution with an overpotential of 351 mV at 10 mA cm^−2^. Furthermore, waste paper-based CAs as electrocatalyst supports enable low-cost waste reuse. The size and microstructure of transition metal electrocatalysts are controlled by rational tailoring hierarchical porous CAs. Song et al. [78] fabricated vertical MoxSy arrays (MoxSy@GCA), with a length of about 100 nm uniformly grown on guar gum-derived CAs (GCA), using an in situ hydrothermal synthesis method (Figure 4g). The layer-by-layer charge transfer paths endow MoxSy@GCA with a large number of active centers and high conductivity. The best Mo_4_S_16_@GCA possesses low onset potentials of HER (24.28 mV) and OER (1.53 V) and has a low overpotential close to noble metal Pt/C at 10 mA cm^−2^, which was HER (54.13 mV) and OER (370 mV).

The performance of CA-based electrocatalysts not only depends on their structural properties but also highly depends on the activity of their active catalytic centers. Peng et al. [82] prepared a novel tubular metal–CA nanocomposite by carbonizing FeCr-doped Ni benzene tricarboxylate aerogels containing encapsulated small-sized individual Fe, Cr, and Ni nanoparticles. Supercritical drying, Cr doping, and the elongated framework of the aerogel alleviate metal aggregation and promote the in situ growth of carbon tubes. This nanocomposite exhibits superior stability and low overpotentials for HER (137 mV) and OER (220 mV), and the cell voltages are as low as 1.54 V at 10 mA cm^−2^. The co-doping of heteroatoms to form multiple active sites and the synergistic effect between heteroatoms can effectively improve the catalytic performance of the catalyst. Lv et al. [79] intercalated NiCoFe alloys into conductive B, N co-doped/biomass-derived CA nanoparticles as OER electrocatalysts, which enhanced OER by increasing the number of electrocatalysts’ active centers (Figure 4h). The best electrocatalyst (BN/CA-NiCoFe-600) exhibits a tiny Tafel slope of 42 mV dec^−1^ and a low overpotential of 321 mV at 10 mA cm^−2^. In addition to the metal-formed active sites, the heteroatom-doped metal-free catalysts also showed good water-splitting ability. Song et al. [80] prepared B, N co-doped CAs, which were synthesized with B, N co-doped carbon nanosheets by a cationic intercalation exfoliation method. This carbon-based bifunctional catalyst (B5/GCS) exhibits excellent HER and OER performance, showing great activity in both acidic and basic media (Figure 4i–m). The low onset potentials of HER and OER in 0.5 M H_2_SO_4_ electrolyte were 39.12 mV and 1.38 V, respectively. Mechanistic studies demonstrate that its excellent water-splitting activity can be attributed to the formation of fragmented nanosheets with topological defects and the synergistic effect of B, N co-doping, which synergistically promotes proton adsorption and catalytic reactions.

Despite the scarcity and high cost of noble metals, noble metals’ platinum and metal oxides such as RuO_2_ and IrO_2_ are still the best catalyst materials for HER and OER. The presence of catalysts at the atomic or sub-nanometer scale metal particles anchored on CAs can effectively maximize atom utilization and enhance electrocatalytic activity. Huang et al. [83] uniformly dispersed Pt on the surface of carbon supports with high density in the form of single atoms or sub-nanometer clusters (average diameter ~0.4 nm). The synthesized Pt_c_/C catalyst (Pt_c_(250)/C) exhibits extremely high Pt mass activity (j_mPt_)/turnover frequencies (TOF) of 154.5 A mg^−1^/156.1 s^−1^ at η = 60 mV, which were ~ 106 times higher than those of commercial Pt/C (1.46 A mg^−1^/1.48 s^−1^). The stability of single-atom catalysts (SACs) is a great challenge for large-scale practical applications due to their highly unsaturated coordination environment. Zhang et al. [84] prepared atomically isolated ruthenium-doped CAs by biomass hydrogel pyrolysis, which exhibited remarkable performance over a wide range of solution pH due to the formation of abundant RuNx moieties. The best sample (NCAG/Ru-3) has ultralow overpotentials at a current density of 10 mA cm^−2^, 65 mV in 0.5 M H_2_SO_4_, 45 mV in 1.0 M phosphate buffer, and 4 mV in −1.0 M KOH, with mass activities 6, 16, and 44 times that of commercial Pt/C, respectively. Combining with other metal-based materials to form alloys or heterostructures is another feasible route for utilizing noble metals. Kobayashi et al. [85] successfully synthesized PtW nanoparticles in a solid solution. Lightly doped with W in Pt nanoparticles, the overpotential reaches 19.4 mV at 10 mA cm^−2^, which was significantly lower than that of commercial Pt/C (26.3 mV). Theoretical calculations show that the negatively charged Pt atoms adjacent to the W atoms provide favorable hydrogen adsorption energy, thereby significantly enhancing the activity of HER. Ru-based catalysts of platinum group metals have been used as alternatives to Pt-based materials due to their low cost (about 1/30 of Pt), Pt-like activity, and excellent stability. Tang et al. [86] prepared Ru aerogels for HER, RuCo aerogels for OER, and combined catalysts for bulk water splitting. The active centers of Ru aerogels are metallic Ru species, which have comparable HER activity and better stability than Pt/C. The obtained Ru_0.7_Co_0.3_ aerogel has the best OER performance over the RuO_2_ benchmark catalyst. Its excellent OER performance is mainly attributed to abundant oxygen vacancies, the synergistic catalytic effect of RuCo, and the structural advantages of the sample. In the actual bulk water splitting test, the combination of Ru and Ru_0.7_Co_0.3_ aerogel catalysts outperformed the Pt/C + RuO_2_ catalysts.

### 2.3. Electrocatalytic CO_2_RR

The CO_2_ resource utilization technology can convert CO_2_ as a raw material into methanol, methane, carbon monoxide, ethylene, and other high-value products, and is considered one of the reliable technologies to achieve the goal of carbon neutrality. Electrocatalytic CO_2_ reduction, driven by renewable energy sources such as wind, solar, and geothermal energy, provides a promising approach to the carbon cycle from chemicals and fuels. However, the chemical structure of the CO_2_ molecule is extremely stable, and it needs to overcome considerable kinetic and thermodynamic energy barriers to complete the reduction of CO_2_ [87,88,89,90]. In addition, the CO_2_ reduction process is often accompanied by the occurrence of hydrogen evolution side reactions and the low selectivity of a specific product. Designing catalysts with high selectivity, suppressing the side reaction of hydrogen evolution, and achieving high current density at low overpotentials is the main focus of researchers in the CO_2_ conversion field. Au and Ag are considered to be the most efficient electrochemical catalysts among all catalyst materials, but their high costs limit their applications. Abundant non-noble metal and nanostructured carbon materials have been investigated to replace Ag- and Au-based catalysts (Figure 5 lists several examples). 

In the early stages of CO_2_RR research, researchers mainly focused on the screening of bulk metals, especially noble metals such as Au and Pt, as catalysts. As the investigation evolved, researchers concentrated more on carbon-based materials and transition metals that are widely available, and inexpensive CAs have been gradually applied to CO_2_RR due to their advantages of good chemical stability, controllable structure, good electrical conductivity, and wide source of raw materials. The higher electrochemically active areas and electronic conductivity lead to faster electron transfer in the CO_2_RR process to form critical intermediates. The introduction of heteroatoms creates defects that affect the overall charge state of the carbon framework and enhances the activity and density of potential active sites, thus improving the electrocatalytic performance of carbon nanomaterials. The N doping method is often used to enhance the electrochemical activity of CO_2_RR catalysts, due to the fact that the N atoms have high spin density and electronegativity-induced charge delocalization. Han et al. [94] used N, P co-doping to change the local charge and spin density of N-doped carbons to obtain N, P co-doped CAs catalysts (NPCA) because P has a larger atomic size and lower electronegativity than N. NPCA has excellent performance for the electroreduction of CO_2_ to CO, and the current density of CO_2_ was as high as −143.6 mA cm^−2^ at a Faradaic efficiency (FE) of 99.1%. Detailed studies show that pyridine N can enhance the activity of CO_2_RR while phosphorus can hinder the adsorption of *H, thus contributing to the high selectivity of CO_2_RR.

The transition metal supported on CA materials has also been developed as a variety of catalysts with high activity and selectivity for CO_2_RR. Lv et al. [91] designed Cu–Zn-incorporated silk fibroin biomass CAs for electrocatalytic CO_2_RR (Figure 5a). Zn nitrate was introduced as a pore former to optimize the pore structure further. The as-prepared SF−Cu/CA−1 catalyst possesses abundant mesoporous structure and unique composition, which facilitates rapid electron transfer, sufficient exposure of active sites, and desorption of *CO intermediates. The electrocatalytic CO_2_RR of the SF−Cu/CA−1 catalyst achieved a FE of CO at 83.06% and the CO/H_2_ ratio of 19.58 at the current density of 29.4 mA cm^−2^. CAs decorated with various active components are reported to suppress the HER greatly in CO_2_RR cells. It is found that N-doped CAs have a specific inhibitory effect on the HER. He et al. [92] synthesized an Mn-based heterogeneous catalyst by depositing MnO nanoparticles on 3D N-doped GAs (NGAs), as shown in Figure 5b. The low mass transfer and enhanced electrical conductivity resistance of N-doped 3D GAs enable HER to be effectively suppressed. The catalyst achieves excellent CO_2_RR performance at −0.82 V vs. RHE with a FE of 86% for CO and better stability for at least 10 h. The introduction of metal single atoms (Cu, Ni, Mn, etc.) on the surface of CAs can change the electronic structure of carbon nanomaterials. Moreover, the introduction of single atoms increases the number and activity of active centers which can better stabilize the intermediate products of CO_2_ reduction and improve the performance of electrocatalytic CO_2_RR. Designing the nanostructure of catalysts to expose more active centers can effectively enhance the catalytic efficiency of the catalyst. Huang et al. [93] designed the Ni/Zn bimetallic imidazolium zeolite framework (Ni/Zn−ZIF−8) and carboxymethyl cellulose composite and carbonized it to obtain a N-doped CA-supported Ni single-atom catalyst (Figure 5c). The hierarchically porous structure makes the Ni single active site on the catalyst surface easily accessible to the electrolyte and CO_2_ molecules. A high CO FE of 95.6%, an industrial-grade CO fractional current density of 226 mA cm^−2^, and a significant turnover frequency of 271,810 h^−1^ were achieved in a flow cell reactor at −1.0 V vs. RHE (Figure 5d–g). This design improves on the disadvantage that the single-atom active centers on the microporous carbon support are not fully exposed, resulting in low current density.

## 3. Summary and Perspectives

Three-dimensional interconnected porous CAs, known as a bridge between nanoscale to macroscale applications, are favored by researchers because of their high porosity, low density, large specific surface area, designable structural, and chemical properties. This paper reviews the recent advances in the application of CAs in electrocatalytic ORR, OER, HER, and CO_2_RR. For electrocatalysis, they can be used as catalysts themselves and as catalyst supports for metals and metal oxides to prevent the reduction in the number of active centers caused by the aggregation and dissolution of metal particles, which leads to the decrease in catalytic performance. The large specific surface area and 3D porous framework of CAs provide abundant active sites and transport channels for electrolytes and reactants, facilitating the proton–electron transfer process. Exposing more active sites is helpful for the improvement in catalytic performance. In addition, incorporating heteroatoms into CAs will increase the rate of carrier transfer on the surface of CAs and improve their electrical conductivity. The binding energy between the catalyst surface and the intermediate product is also reduced, which is beneficial to improving the catalyst performance. Furthermore, the synergistic effect between heteroatoms and metal nanoparticles enhances the catalytic activity of the catalysts.

There are several challenges in applying CAs to industrial-scale electrocatalytic production with current technologies. First of all, the complex, high-cost synthesis process of CAs and the difficulty in adjusting the pore structure of CAs are factors affecting the industrialization of CAs. The commonly used synthesis processes are supercritical drying and freeze-drying, and the drying process requires a large amount of energy. Furthermore, the existing technology cannot freely adjust the structure of the hole in a wide range. Secondly, the poor mechanical properties of CAs are limited in industrial production. Most of the current CA electrocatalysts still use the powder state for reaction. The powdered catalyst has the disadvantages of difficult recovery and easy aggregation and deactivation. Finally, most studies on the synthesis of CA-based catalysts are still being screened one by one experimentally, which is time-consuming and labor-intensive at present.

We should focus on developing a class of facile and efficient solid bulk CA-based electrocatalysts. The first priority is to develop a low-cost (such as raw material selection of biomass materials), simple, fast, and mass-produced CA preparation method. The second is to develop strong solid bulk CAs. The CAs are complexed with other substances and different molecules are added to improve the mechanical stability and adsorption capacity of CAs. The third priority is to use theoretical predictions or modelling to assist in the selection of catalysts. Its application reduces the material and time cost of experimental trial and error and starts from the mechanism to obtain a more in-depth study.

## Figures and Tables

**Figure 1 nanomaterials-12-02721-f001:**
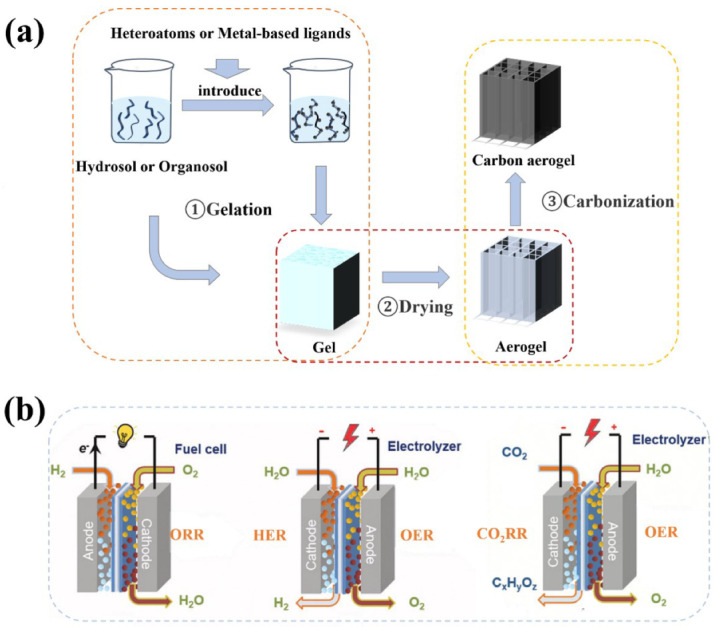
(**a**) Schematic diagram of the synthesis process of CAs. (**b**) The electrochemical processes include ORR in fuel cells, the OER and HER in water electrolyzers, and the CO_2_RR in CO_2_ reduction electrolyzers [33]. Reproduced with permission. Copyright 2021, Wiley-VCH.

**Figure 2 nanomaterials-12-02721-f002:**
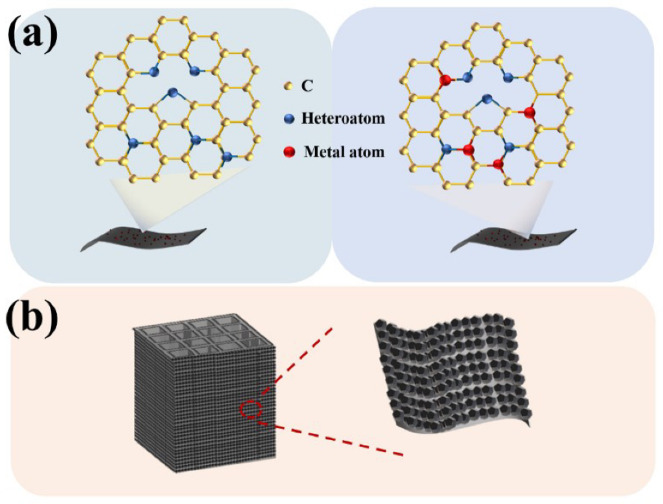
(**a**) Heteroatom-doped and metal-doped carbon structures. (**b**) Hierarchical porous structure assembled by CAs as carriers.

**Figure 3 nanomaterials-12-02721-f003:**
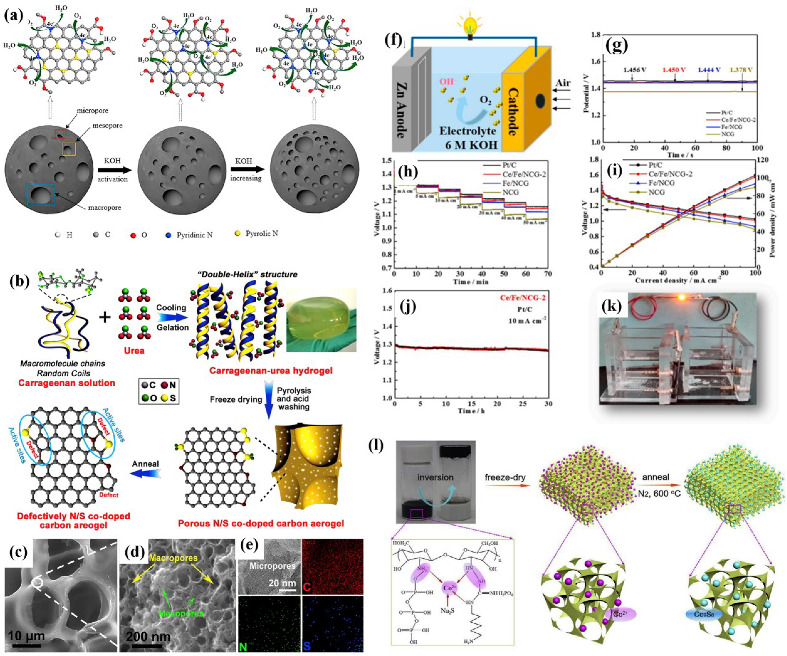
(**a**) Schematic illustration of possible ORR mechanism for CA and CA−KOH [50]. Reproduced with permission. Copyright 2018, Elsevier. (**b**) Schematic illustration of the synthesis process. (**c**,**d**) Low-resolution (**c**) and high-resolution (**d**) FESEM images of NSCA−700−1000. (**e**) TEM image and the corresponding EDS mapping for C, N, and S elements of NSCA−700−1000 [51]. Reproduced with permission. Copyright 2018, Elsevier. (**f**) Schematic illustration of zinc–air battery. (**g**) Open circuit voltage, (**h**) Cell voltage vs. time at various current densities, and (**i**) Polarization and power density curves of samples NCG, Fe/NCG, Ce/Fe/NCG−2 cell, and (**j**) Pt/C cell with 6 M KOH electrolyte, long-term galvanostatic discharge of Ce/Fe/NCG−2 and Pt/C cell, and (**k**) assembly illustration of zinc–air battery [52]. Reproduced with permission. Copyright 2021, Elsevier. (**l**) Schematic illustration of the preparation of Co_9_S_8_/N and P-APC [53]. Reproduced with permission. Copyright 2019, Elsevier.

**Figure 4 nanomaterials-12-02721-f004:**
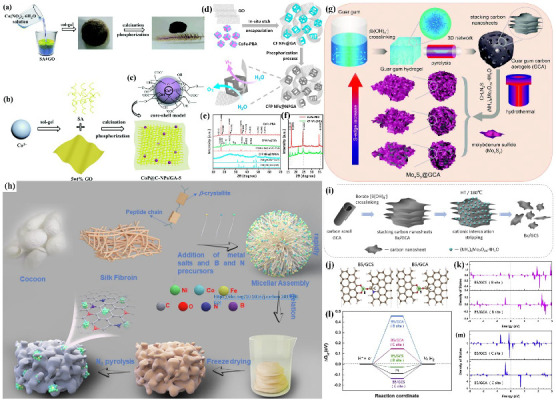
Schematic description of facile synthesis. (**a**) The fabrication of CoP@C−NPs/GA-x obtained from SA/graphene gelation. (**b**) The synthetic route of CoP@C-NPs/GA-5. (**c**) Core-shell model schematic of CoP@C−NPs [76]. Reproduced with permission. Copyright 2018, The Royal Society of Chemistry. (**d**) Schematic diagram of the preparation of the hollow CFP NFs@NPGA. (**e**) The XRD patterns of CoFe-PBA, CF NFs@GA, and CFP NFs@NPGA. (**f**) The comparison of the crystal phase between CoFe−PBA and CF NFs@GA. Reproduced with permission [77]. Copyright 2020, Elsevier. (**g**) Schematic of the synthesis of MoxSy@GCA [78]. Reproduced with permission. Copyright 2019, The Royal Society of Chemistry (**h**) Schematic illustration of BN/CA−NiCoFe fabrication process. Reproduced with permission [79]. Copyright 2022, Elsevier. (**i**) Schematic illustration of the construction of Bx/GCS. DFT-calculated HER activities of B5/GCS. (**j**) The optimal configuration of B5/GCS and B5/GCA. Green, red, blue, and brown spheres represent B, O, N, and C atoms. (**k**) The projected density of states on B site of B5/GCS and B5/GCA. (**l**) The projected density of states on the C site of B5/GCS and B5/GCA. (**m**) The calculated free-energy diagram of HER at the equilibrium potential for carbon-based catalysts (B5/GCS and B5/GCA with different active sites and Pt reference. Reproduced with permission [80]. Copyright 2019, Elsevier.

**Figure 5 nanomaterials-12-02721-f005:**
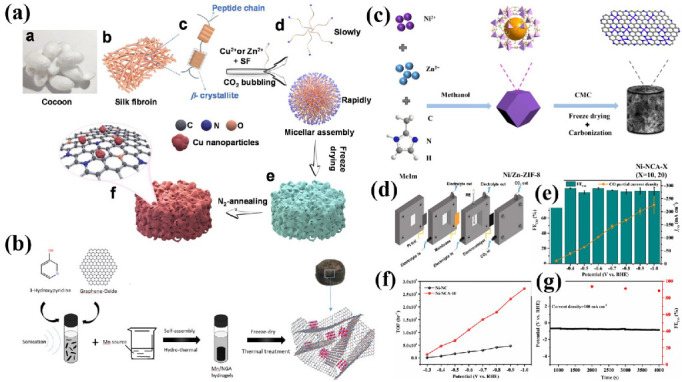
(**a**) Schematic of the synthesis process of the SF-Cu/CA or SF-Cu/CA-1. Reproduced with permission [91]. Copyright 2021, Elsevier. (**b**) Schematic Illustration of the preparation of Mn-based heterogeneous catalysts. Reproduced with permission [92]. Copyright 2021, ACS Publications. (**c**) Schematic illustration of the fabrication of Ni SACs supported on CAs based on the Ni/Zn−ZIF−8/carboxymethylcellulose aerogel. (**d**) Schematic of a flow cell configuration. (**e**) FE_CO_ and j_CO_ of Ni-NCA−10 in the flow cell at different potentials. (**f**) TOF of Ni-NCA-10 compared with Ni−NC in an aqueous solution of 1 M KOH in the flow cell. (**g**) Stability of Ni−NCA−10 at a current density of 100 mA cm^−2^ in the flow cell. Reproduced with permission [93]. Copyright 2022, Elsevier.

**Table 1 nanomaterials-12-02721-t001:** Classification and their properties according to CAs’ precursors. (•, √, * corresponding to the CA obtained from different precursors.)

CAs Categories	Example	Precursor	Carbonization	Gelation	Properties
Traditional CAs	Organic-based CAs	Aromatics (Resorcinol, phenol, phloroglucinol) and aldehydes (formaldehyde, furfural)	Yes	Yes	• Structural properties are controlled by synthetic conditions.
		Polymers (poly (vinyl alcohol), polyimide)			• Suitable for mass production.
Emerging CAs	Graphite-based CAs	Graphene, carbon nanotubes, carbide, carbon nanofibers	No	Yes	√ Potential candidates for conductive materials.
					√ Cross-linking through van der Waals interactions.
	Biomass-based CAs	Natural polysaccharides (chitosan, konjac glucomannan, cellulose, lignin), protein (silk fibroin)	Yes	No	* Wide source, low price, green and environmental protection

## Data Availability

Not applicable.

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
