# Peer review of "Carbon Aerogels as Electrocatalysts for Sustainable Energy Applications: Recent Developments and Prospects"

_nanomaterials, 2022, doi:10.3390/nano12152721_

Round 1

Reviewer 1 Report

The review is dedicated to the application of the carbon aerogels for the electrocatalytic reactions in energy storage processes. The manuscript is well organized and the presented data corresponds modern state of art.

Small remarks:

1. Page 2 "A gel whose liquid phase is water is called a hydrogel. Gels are generally divided into two categories: hydrogels (the liquid phase is a gel of water) and organogel (the liquid phase is a gel of water)." - mistype in description of organogel

2. "organic aerogels such as RF aerogels" - what is RF abbreviation?

3. Page 3. In Table 1 it is difficult to understand whether the last column describes overall properties of the CAs or these properties should be attributed to the CAs produced from different sources

4. "3D interconnected porous nanostructured CAs have been widely used in electrocatalysis due to their excellent electrical conductivities" - please expand this sentence with the data because of conductance of polymers is usually not so high.

 5. Please provide the experimental data about pure carbon CAs in Table 2.

Reviewer 2 Report

The manuscript is focused on the carbon aerogels electrocalatysts for sustainable energy applications. Thus, it could be interesting to review the use of CA as electrocatalysts for oxygen reduction reaction, oxygen evolution reaction, hydrogen evolution reaction and CO2 reduction reaction. Some recommendations to improve the understanding of work follow below.

-Manuscript should be better organized. There are many objectives, but little developed. 

Author Response

Thank you very much for your suggestion. The use of CAs for electrocatalytic reactions is still in a developing stage. The manuscript focuses on reviewing the synthesis, applications, and reaction principles of CA as an electrocatalyst for the oxygen reduction reaction, oxygen evolution reaction and hydrogen evolution reaction, and CO2 reduction reaction. I clarified the logic of the article, summarized the role of CAs in the electrocatalysis process in the order of time development, and indicated the status of CAs in electrocatalysis. I marked the unclear places with symbols. In the summary , I highlighted the development direction of CAs in electrocatalysis and put forward some feasible suggestions for this.

Reviewer 3 Report

The review article „Carbon Aerogels Electrocatalysts for Sustainable Energy Applications:

Recent Developments and Prospects” describes the use of carbon aerogels in multiple electrochemical catalytical processes like oxygen reduction reaction (ORR), oxygen evolution reaction (OER), hydrogen evolution reaction (HER), and CO2 reduction reaction (CO2RR). The authors describe the basics of the processes and literature findings in each case. The article is written thoroughly, the English language is excellent, from my point of view. The figures are prepared clearly.

I have an objection only to the last sentence in the conclusions, “The third is to use machine learning to assist in the selection of catalysts. The application of the machine reduces the material and time cost of experimental trial and error, and starts from the mechanism to obtain a more in-depth study.” I would stay by theoretical predictions or modelling instead of machine learning approaches. The application of the “machine” also doesn’t sound well.

Author Response

This manuscript is a resubmission of an earlier submission. The following is a list of the peer review reports and author responses from that submission.